# Relationship Analogy between Sleep Bruxism and Temporomandibular Disorders in Children: A Narrative Review

**DOI:** 10.3390/children9101466

**Published:** 2022-09-25

**Authors:** Yeon-Hee Lee

**Affiliations:** Department of Orofacial Pain and Oral Medicine, Kyung Hee University School of Dentistry, Kyung Hee University Medical Center, #613 Hoegi-Dong, Dongdaemun-gu, Seoul 02447, Korea; omod0209@gmail.com; Tel.: +82-2-958-9454; Fax: +82-2-968-0588

**Keywords:** sleep bruxism, children, temporomandibular disorder, obstructive sleep apnea, rhythmic masticatory muscle activity, non-REM sleep

## Abstract

Sleep bruxism (SB) is a condition characterized by repetitive clenching or grinding teeth and/or by bracing or thrusting of the mandible during sleep. Although SB is not considered a disorder in children, SB can be a potential physical and psychological hazard or consequence, and this study examines whether SB is a risk factor for TMD as it is in adults. A narrative review on the topic of inferring the relationship between sleep bruxism and TMD in children was performed based on a search in the PubMed and Google Scholar databases for articles published between 1999 and 2022. A total of 76 articles were included in this review. SB is very common in children, may be related to psychological distress or sleep breathing disorder, with a prevalence of up to 49%, and mainly occurs in the non-rapid eye movement stage in the sleep structure. SB may be one cause of TMD. The prevalence of TMD in children is 16–33%. Compared to the female-dominant TMD prevalence in adults, the sex-differences in TMD prevalence in children are less pronounced. However, TMD-related pain is more prevalent in girls than in boys. Given the complex etiology of each of SB and TMD in children, it can be inferred that the explanation of the relationship between the two conditions is very challenging. Ultimately, their relationship should be understood in the individual biopsychosocial model in the process of special physical growth and mental development of children. Moreover, appropriate clinical guidelines for a definitive diagnosis of SB and TMD in children and more research with a high scientific evidence level, which is comprehensive, considering physical, psychological, genetic, and social cultural factors, are required.

## 1. Introduction

In general, bruxism is a repetitive and nonfunctional activity of masticatory muscles characterized by clenching or grinding of teeth. In 2013, an international consensus was reached on a simple and practical definition of bruxism as repetitive masticatory muscle activity characterized by clenching or grinding teeth and/or by bracing or thrusting of the mandible [1], and largely divided into awake bruxism and sleep bruxism (SB) according to the circadian phenotype [1,2]. According to the International Classification of Sleep 3, the criteria for SB classification include the presence of regular or frequent tooth grinding sounds during sleep and one or more of the following clinical signs and symptoms: morning jaw locking as a result of tooth grinding during sleep, transient morning masticatory muscle pain or fatigue, temporal headache, and/or severe tooth wear [3]. The American Academy of Sleep Medicine defines SB as a movement disorder characterized by the clenching and grinding of teeth during sleep [4]. SB is considered the most frequent form of parasomnia in children. In addition, SB is considered a potential risk factor for temporomandibular disorders (TMDs) [5]. SB is not considered a disorder in children. However, this study aims to examine the potential physical and psychological risks of SB in children and whether SB is a hazard for TMD as in adults. In addition, this study investigated whether SB in children is a result of psychological stress, sleep disturbance, and sleep disturbance-breathing in children, and whether SB is a risk factor for clinical outcomes such as tooth and temporomandibular joint damage and TMD pain. Given the importance of SB and TMD in children, it has not been reviewed recently. Therefore, the present study will discuss the diagnosis, etiology, pathophysiology, and complications of SB, and the relationship between SB and TMDs in children was of particular interest.

## 2. Materials and Methods

A narrative review was performed based on a search of PubMed and Google Scholar databases for articles on SB in children and their relationship with TMDs. Keywords used in the search to find related articles are: “children”, “bruxism”, “sleep bruxism”, “grinding”, “clenching”, “temporomandibular joint (TMJ)”, “masticatory muscle”, “orofacial”, and “TMD”. Papers published in English between January 1999 and March 2022 were filtered in this search. In scientific research and biology, children are generally defined as people between birth and puberty.

Early childhood lasts from birth to approximately 7 years of age, and middle childhood begins at approximately 7 years of age and usually ends with puberty (around 12 or 14 years of age), which marks the beginning of adolescence [6]. Thus, in this study, children were referred to as up to 14 years of age. Studies that partially or completely included patients or participants in this age group were reviewed. Adolescence mostly refers to the age of 13–19 years and corresponds to the post-childhood stage.

Articles were preselected after review of titles and/or abstracts, and articles that did not meet the inclusion criteria were excluded. Data extracted from the study included study characteristics, participants, and outcomes. Prior inclusion criteria for literature review were: randomized clinical trials (RCT), and observational studies of diagnosis, prevalence, etiology, risk factors, and the relationship between SB and TMD. For original research, articles were included when the following conditions were satisfied; definite sleep bruxism has been diagnosed with objective measurement or instrumental assessment, with or without a positive self-report and/or a positive clinical inspection, and TMD has been diagnosed using the research diagnostic criteria for TMD (RDC/TMD) or diagnostic criteria for TMD (DC/TMD) protocol or the equivalent objective diagnostic method. RDC/TMD and its 2014 updated version, DC/TMD, are reliable and valid diagnostic tools of TMD [7,8]. The following were excluded from the literature review: articles published in languages other than English, simple case reports, and low-quality reports with risk of bias.

A total of 1359 articles were retrieved from the PubMed and Google Scholar databases from January 1999 to March 2022 (Figure 1). After applying the inclusion and exclusion criteria to all articles, and analyzing the abstracts and full texts of some articles, 76 articles were finally selected. The author reviewed repeatedly over a 2-week period and mainly tried to verify the content and study design to determine whether the paper was suitable for this study.

## 3. Results

### 3.1. Etiology of Sleep Bruxism

Although bruxism has an uncertain or controversial etiology, it is considered complex and multifactorial [9]. The etiology of bruxism requires physical, psychological, hereditary, and genetic factors. In children, physical factors of the oral cavity, including nasal obstruction, tonsil hypertrophy, and restricted tongue mobility, may have a synergistic relationship with the occurrence of SB [10]. Furthermore, psychological disturbances appear to be a crucial cause of sleep problems. Psychological problems, including anxiety and stress, are strongly associated with bruxism [11]. Children with anxiety are common in pediatric psychiatry. Somatic anxiety can increase the muscle tone. A previous study using polysomnography (PSG) found an increased incidence of SB in children with tension-type headaches [12]. Interestingly, psychological distress and anxiety are also reported as a common risk factor for TMD [13]. In addition, central mechanisms involving brain neurotransmitters are also important in the onset and development of SB. Bruxism is primarily controlled by the central nervous system and might be related to central nervous system hyperexcitability due to disturbances in the GABAergic and glutamatergic systems of the brain [14]. Neuroinflammation and hormonal disturbances may be involved in stress and SB [15]. However, it is controversial whether the socioeconomic status of children affects the occurrence of SB [16]. Genetic factors, such as those related to serotoninergic neurotransmission, have been implicated in both psychological and systemic conditions and may contribute to the cause of SB [17]. Finally, depending on etiopathogenesis, SB can be divided into primary (idiopathic) and secondary (associated with diseases or specific medications) [18]. Certain mental and medical disorders including epilepsy, dementia, night terrors, and Parkinson’s disease are related to bruxism. Certain types of drugs and chemicals can increase the frequency or intensity of SB episodes. The most commonly mentioned compounds are selective serotonin reuptake inhibitors, selective norepinephrine reuptake inhibitors, antipsychotics, 3,4-methylphenidate (ecstasy), amphetamines, nicotine, and alcohol [19]. Certain drugs can also interfere with the normal secretion and function of neurotransmitters in the central nervous system to induce bruxism in children [20].

### 3.2. Prevalence of Sleep Bruxism in Children

The prevalence of bruxism was observed in all the age groups. Regularly occurring SB was reported in 8.6% of the general population and decreased with age [21]. In young adults aged between 18 and 29 years, it is 13%, reduced to 3% in individuals over 60 years of age [22]. In adults, a prevalence of 5.0% was found for awake bruxism, and 16.5% for SB [23]. The prevalence of SB in children is higher than in adults, reaching up to 49% [24,25]. Epidemiological studies have been conducted with various methodologies and populations, and for this reason, the prevalence of SB may vary. SB was diagnosed in 56.5% of girls and 43.5% of boys, with a higher prevalence in girls [26]. In addition, among adolescents aged 11–14, the incidence of SB was higher in adolescent males than in adolescent females, and the incidence rate was 22.2% [27]. Considering the age group, further investigation is needed to determine whether there is a difference in the prevalence of SB according to sex.

### 3.3. Sleep Structure and Sleep Bruxism

Considering the sleep structure, SB occurs in the light sleep stage of non-rapid eye movement (REM) sleep in both adults and children. Sleep can be divided into 3–5 cycles, usually consisting of one cycle with non-REM and REM periods ranging from 90 to 120 min. Non- REM sleep can be further divided into stages 1 and 2, which are light sleep, and stages 3 and 4, which are deep sleep. SB episodes mainly occur during the mild stages of non-REM sleep, but occasionally (<10%) also occur in REM sleep, and are thought to be associated with sleep arousal [28]. SB in REM sleep is characterized by momentary cortical brain activation and increases in motor activity and heart rate [29]. During REM sleep, human brain activity is similar to the level experienced when awake, but muscles are usually maximally relaxed. In children, SB occurs in both non-REM and REM sleep (stage 2) [30]. However, since polysomnography in children has mostly been limited, more research on sleep structure is needed.

### 3.4. Rhythmic Masticatory Muscle Activity (RMMA) and Sleep Bruxism

During sleep, RMMA is more common in patients with SB (80%) than normal subjects (60%). RMMA is slow with 1-Hz chewing-like mandibular movements, and SB can be identified when RMMA is frequent. A serial physical change over time related to micro-arousals has been suggested as the mechanism of RMMA and SB (Figure 2) [31]. RMMA is up to three times more common and about 30% more intense in SB patients than in normal controls [32]. A physiological relationship between sleep bruxism and RMMA has not yet been established. As RMMA may be coupled with increases in salivation to lubricate the oral cavity and oropharyngeal structures and to enlarge upper airway spaces, RMMA may be a normal activity to maintain homeostasis of the oral cavity and the deep respiratory system. SB and RMMA may be associated with the central pattern generator, which is located in the trigeminal nucleus [33]. A central pattern generator controls the rhythmic masticatory behavior when a person is awake.

### 3.5. Diagnosis of Sleep Bruxism in Children

The clinical evaluation and definitive diagnosis of bruxism are usually challenging [2]. To accomplish this, various tests such as subjective observation by the patient, observation by family members or sleep partner, clinical examination, electromyography (EMG), evaluation using an oral device, and PSG may be required. [34]. In adults, self-reported SB, sleep grinding referral by a bedpartner, and objective clinical assessment are important for diagnosis [35,36]. Diagnosis of SB in children is mostly based on reports of non-children, family members, and most often of parents, that describe the characteristic sounds of teeth grinding during sleep. If the clinician interviewed the child and comprehensively evaluated the questionnaires that the parents answered, the effectiveness of SB could be trusted [37,38]. However, definitive diagnosis of SB requires reporting of clinical status consistent with bruxism and confirmation of PSG, which measures EMG activity of masticatory muscles associated with bruxism during sleep testing by audio and video recordings [39]. The clinical diagnostic criteria for SB are grinding or clenching teeth during sleep, attrition of one or more abnormal teeth, tooth sounds associated with bruxism, and pain or discomfort in the masticatory muscles [40]. For children, it is not easy to recognize SB on their own, and the most reliable clinical method for diagnosing SB in children is for parents or caregivers to report their bruxism. In addition, since parents do not always recognize or observe their children’s tooth clenching and/or grinding, an objective examination is necessary for an accurate and definite diagnosis of SB.PSG is considered the gold standard for diagnosing SB. However, the use of PSG in large samples is still not feasible owing to the high cost and need for qualified specialists [41]. Recently, interesting mobile and consumer wearable devices for detecting sleep breathing disorders have been developed [42,43]. The combination of these techniques with the EMG channel may facilitate the objective diagnosis of SB in children. Although it is still an abstract approach, a non-invasive diagnosis may be attempted by measuring the decreased salivary cortisol level after awakening, which is related to sleep disturbance [44]. Currently, validated and reliable criteria for the SB diagnosis in children are still lacking. Due to the ambiguity of the diagnosis of SB in children, the treatment is also not well established, so standardization and development of diagnostic criteria for SB is desperately needed [45]. It should be emphasized that, unlike adults, early diagnosis without diagnostic relay and identification of risk factors in children have an important meaning as they can inhibit craniofacial skeletal changes, help relieve orofacial and TMD pain, and prevent destruction of teeth or dental restorations.

### 3.6. General Complications of Sleep Bruxism in Children

SB can interfere with a restful night of sleep, leading to fatigue and daytime sleepiness the following day. SB-related parameters trigger the release of catecholamines in the central nervous system, influencing the release of chemical mediators that alter arousal and the initiation and maintenance of sleep [46]. However, people with SB are surprisingly generally unaware of the harmful factors that can eventually lead to abnormal tooth wear, generalized attrition and cervical abrasion of the teeth, abnormal tooth wear, gingival recession, and the development of TMD (Figure 3). In general, bruxism can cause pathological tooth and periodontal tissue destruction, damage to prosthetic reconstruction, failure of dental procedures, TMJ and masticatory muscle pain, mandibular mobility limitation, masticatory muscle pain and fatigue, muscle hypertrophy, and headache [47]. Conversely, Smardz et al. reported that there was no significant correlation between the intensity of sleep bruxism and TMD-related pain [48]. When participants were divided into sleep-bruxers and non-bruxers, the distribution of TMD did not differ statistically significantly [49]. In children, SB can also be associated with TMD symptoms and may have additional symptoms, such as ear pain and other muscle pain [50,51]. In addition, SB can co-occur with other sleep disorders, which may be associated with increased muscle activity, abnormal body movements such as restless reg syndrome, breathing difficulties, and heart rate disturbances [28]. Therefore, SB can impair the children’s quality of life or learning ability [51]. Unfortunately, children may not clearly report symptoms of bruxism to parents or caregivers; conversely, parents cannot monitor their children for bruxism on a daily basis [52]. Furthermore, parents and caregivers may have difficulty accessing appropriate treatment and management for children due to their insufficient knowledge of the potential or direct effects of SB, thus contributing to the persistence of bruxism in children, which can lead to complications in adulthood [52]. It should be emphasized that studies on the adverse effects of SB in children have been very limited so far, and in particular, observational studies for a long period of time are needed to examine the effects of bruxism occurring in children into adults.

### 3.7. TMD in Children

TMD is an umbrella term that encompasses clinical pain and dysfunction involving the TMJ, masticatory muscles, and their associated structures [53]. Myofascial pain, disc displacements, joint pain, and degenerative and inflammatory joint diseases are the major and common subtypes [54]. Common signs and symptoms are as follows: TMJ sounds, limitation of mouth opening, preauricular pain, restricted mandibular movement, muscle tenderness or pain on mandibular function, headaches, and sleep problems. Both physical and psychological axes that negatively affect the chewing system are considered risk or exacerbating factors for TMD [55]. Overt macrotrauma-producing injuries to the head and neck and jaw, and microtrauma with parafunctional habits can result in prolonged TMD signs and symptoms [56]. In addition, hormonal, immunological, genetic, and psychosocial factors are also considered initiating and/or perpetuating factors for TMD. Epidemiological studies have shown that signs and symptoms of TMD can be found at any age [57]. Previous studies investigating the occurrence of TMD in children and adolescents are limited, and fewer studies have used standardized criteria for the diagnosis of TMD. [58,59].

Interestingly, the signs and symptoms of TMD generally increase with age [60]. The prevalence of TMD is 16–33% in children [8,61]. The varying prevalence may be related to the fact that most signs and symptoms in young children are difficult to detect because most signs and symptoms are characterized as mild, and severe dysfunction is rare. The lack of a globally agreed diagnostic tool of TMD for children also contributes to its wide prevalence. DC/TMD for TMD diagnosis is for adults 18 years of age or older, but fortunately, DC/TMD for children and adolescents is currently being developed [7,59]. Although sex differences in the incidence of signs and symptoms were small in children, TMD was 1.5–2 times more prevalent in women than in men after late adolescence [58].

### 3.8. Relationship between Sleep Bruxism and TMD in Childhood

SB in children may be a comorbid condition for TMD. Sustained clenching and bruxing are the most detrimental activities of the TMJ and masticatory muscles, producing an overload that could lead to severe damage to this tissue [62]. That is, repetitive overload during SB can lead to TMJ disc displacement, degenerative changes of the TMJ, muscular inflammation, and muscle hypertrophy. In addition, SB can also be associated with common TMD symptoms, such as TMJ pain and dysfunction, as well as additional symptoms, such as ear pain and other muscle pain [50,51,63]. Frequent headaches, difficulty in opening or closing the mouth, and ear pain can be accompanied by bruxism and TMD. In some cases, bruxism can lead to the onset of TMD or aggravate existing TMD condition. In recent systemic review, children with bruxism were 2.97 times more likely to have TMD than children without bruxism [64]. However, SB is a multifaceted motor behavior, which should be evaluated on a continuum spectrum rather than using a simplified dichotomous approach of presence/absence, and this complexity should also be considered in relation to TMD [65]. In addition, co-factors of SB and TMD include psychological stress and depression, which increase the severity of SB and TMD symptoms [66,67]. TMD can be caused by stress or related SB.

Furthermore, pediatric sleep-disordered breathing should be considered in TMD in children. The relationship between sleep disturbances and TMD has been established to be bidirectional. In adults, the most common sleep problem accompanying TMD was SB (67%), followed by insomnia (37%), and obstructive sleep apnea (OSA) (23.3%) [68]. However, the detection and diagnosis of pediatric OSA has not been fully elucidated, so the prevalence is estimated to have a wide range, from 0.1 to 13.0% [69]. With SB and OSA being co-occurring sleep conditions, OSA may give rise to episodes of teeth grinding [70]. In patients with OSA, SB may be a reflex action triggered by the brain to resolve the pauses in breath. Similar to the occurrence of SB in sleep structures, OSA can occur in both non-REM and REM sleep [71]. These sleep problems can adversely affect TMD pain by causing aggravation of the systemic inflammatory response and lack of oxygen supply. In addition, OSA may increase the risk of psychological deterioration in children, especially depressive disorder [72]. Therefore, additional studies on pediatric sleep-disordered breathing are needed to clarify the relationship between SB and TMD in children.

Conversely, bruxism may also be a coping strategy of psychological stress [73]. Both SB and psychological distress can be risk factors for TMD. Therefore, SB and TMD may be in an ironic relationship, as mental derangement adversely affects the onset, progression, and aggravation of TMD [74]. In addition, there is no clear cutoff point for bruxism’s influence that divides it into neutral, beneficial, or pathological behavior [2]. SB may have a positive role in lubricating the oral cavity and oropharynx by stimulating salivary secretion and increasing airway patency [28]. In fact, scientific evidence to clarify the relationship between SB and TMD in children is still lacking. In adolescents, bruxism is considered a self-limiting condition that does not progress to adult bruxism and does not appear to be associated with TMJ symptoms [75]. In children and adolescents aged 10- to 18-years old, anterior tooth wear was not related to self-reported TMD pain [76]. Thus, some researchers argue that the relationship between SB and TMD remains controversial.

### 3.9. Study Limitations

There is insufficient, statistically relevant data to draw certain conclusions about the relationship between SB and TMD in children, and there is a lack of multi-center studies with controlled clinical trials or RCTs on this issue. Due to the lack of original articles accompanying objective diagnostic tools and reliable evaluation method for SB or TMD, the precise role of SB in TMD, conversely TMD in SB, still requires further clarification. However, this review focuses on the etiology, diagnosis, clinical characteristics, and related complications of SB in children. A close association could be inferred between SB and TMD in children. Further research is needed in pediatric dentistry or orofacial pain fields to elucidate the relationship between SB and TMD.

## 4. Conclusions

From analogies from previous studies, a relationship between SB and TMD in children can have suggested. SB can cause tooth wear, masticatory muscle pain, TMD pain, limited mouth opening, ear pain, and headache. However, since the diagnostic criteria for SB and TMD in children have not been clearly established, it is difficult to make recommendations for their diagnosis, prevalence estimation, and relationships. Some researchers argue that the relationship between SB and TMD has recently appeared to be controversial and unclear. To identify the relationship between SB and TMD in children, more research based on the research design of high scientific levels, which is comprehensive and considers physical, psychological, genetic, and social cultural factors, is required in the future.

## Figures and Tables

**Figure 1 children-09-01466-f001:**
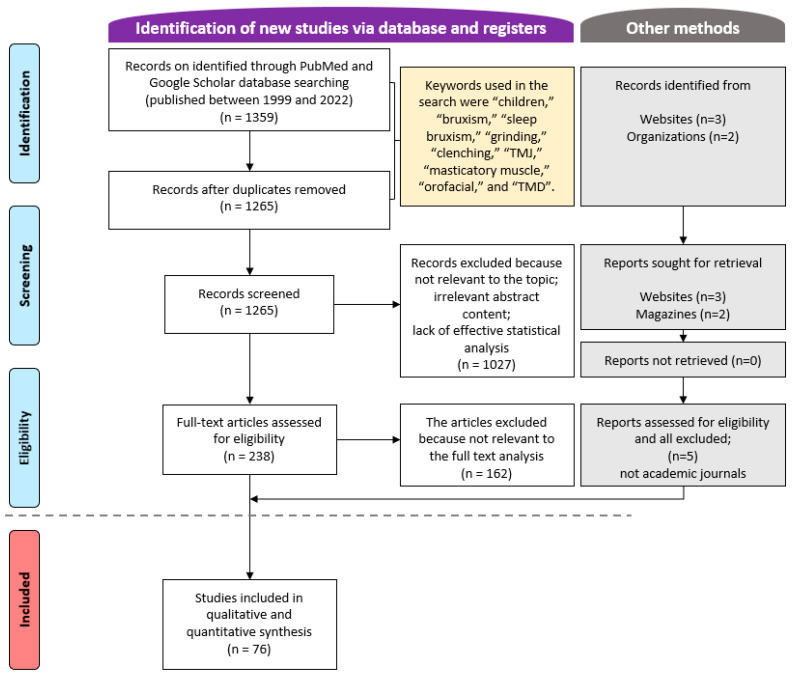
Flow chart of the article selection process.

**Figure 2 children-09-01466-f002:**
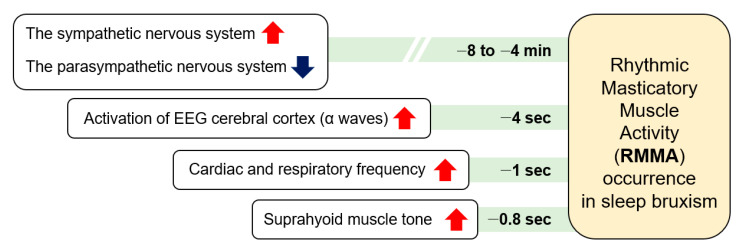
The serial occurrence of rhythmic masticatory muscle activity and sleep bruxism.

**Figure 3 children-09-01466-f003:**
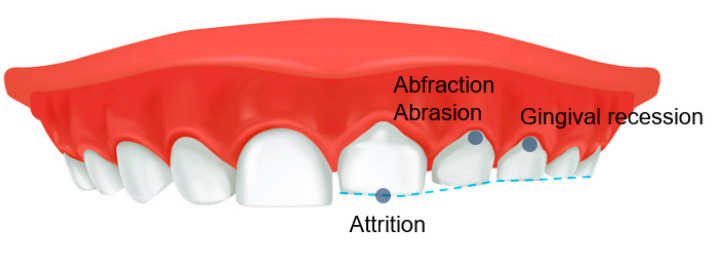
Adverse effects of sleep bruxism on teeth in children.

## Data Availability

The data that support the findings of this study are available from the corresponding author, Y-.H.L. upon reasonable request.

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
