# Peer review of "Relationship Analogy between Sleep Bruxism and Temporomandibular Disorders in Children: A Narrative Review"

_children, 2022, doi:10.3390/children9101466_

Round 1

Reviewer 1 Report

We read with great interest the manuscript with title “The Association Between Sleep Bruxism and Temporomandibular Disorders in Children: A Literature Review

 aiming to describe the association of sleep bruxism and TMD in children.

The work is of interest; however, several criticisms have been found and need to be addressed before resubmission.

1.     Please add to the abstract the aim of the study.

2.     You are asked to carefully check the spelling in the text, see for example line 76. 

3.     You are asked to enter either in the flow diagram/flow chart or in the text (M&M) the search string used on PubMed and the one used on Scholar.

4.     Please use the updated version of the Flow Chart.

5.     Please add to the Flow Chart the reasons for exclusion of the studies.

6.     Please add this citation in the discussion in order to improve the manuscript text and readability: doi: 10.1080/08869634.2019.1581470

7.     M&M: please add the definition of the “children”, as the age range as criteria to include /exclude the studies; and the clinical condition (you included only studies with healthy patients or patients who had comorbidities?).

8.     Results: 63 studies were included; however, it is needed to add information about these studies. You are asked to add:

-the setting of the included studies (hospital, sleep clinic, dental clinic, university hospital clinic, etc..)

-total number of subjects involved (please summarize the total number of subjects being described in the 63 included studies)

-mean age of the subjects

-studies design (case control, cross-sectional, ect..)

-years of publication 

-main outcome

-criteria to diagnose TMD

-criteria used to diagnose SB

-country of the study

In my opinion you should provide a table with the characteristics of the studies.

9.     In order to be consistent with the title of the study being focused on children, I think it would be better to provide an image of a child teeth/mouth instead of figure 3 that represents an adult setting.

Thank you.

Author Response

Reviewer 1

Open Review

English language and style

( ) Extensive editing of English language and style required
( ) Moderate English changes required
(x) English language and style are fine/minor spell check required
( ) I don't feel qualified to judge about the English language and style

Is the work a significant contribution to the field?

Is the work well organized and comprehensively described?

Is the work scientifically sound and not misleading?

Are there appropriate and adequate references to related and previous work?

Is the English used correct and readable?

Comments and Suggestions for Authors

We read with great interest the manuscript with title “The Association Between Sleep Bruxism and Temporomandibular Disorders in Children: A Literature Review”

 aiming to describe the association of sleep bruxism and TMD in children.

The work is of interest; however, several criticisms have been found and need to be addressed before resubmission.

Response:

I sincerely appreciate your constructive comments and suggestions. I did the best I could in the time given. Please refer to the part that has been corrected or added in red. Thank you sincerely.

  1. Please add to the abstract the aim of the study.

Response:

Thank you very much for your constructive and positive comment. The goal of the study was added.

  1. You are asked to carefully check the spelling in the text, see for example line 76. 

Response:

Thank you for your comments.

  1. You are asked to enter either in the flow diagram/flow chart or in the text (M&M) the search string used on PubMed and the one used on Scholar.

Response:

Based on your suggestions, we moved the flow chart to the M&M area. Thank you

  1. Please use the updated version of the Flow Chart.

Response:

I modified the flow chart using the updated version in 2021. Thank you very much for your kind comment.

  1. Please add to the Flow Chart the reasons for exclusion of the studies.

Response:

The reasons why journals were excluded were written and added to the flow chart.

  1. Please add this citation in the discussion in order to improve the manuscript text and readability: doi: 10.1080/08869634.2019.1581470

Response:

Thank you for your comments. Added the reference you provided.

  1. M&M: please add the definition of the “children”, as the age range as criteria to include /exclude the studies; and the clinical condition (you included only studies with healthy patients or patients who had comorbidities?).

Response:

Thanks for your constructive comments. In addition, the inclusion criteria have been clarified. In this study, I added the definition and scope of children.

Early childhood lasts from birth to approximately 7 years of age, and middle childhood begins at approximately 7 years of age and usually ends with puberty (around 12 or 14 years of age), which marks the beginning of adolescence [1]. Thus, in this study, children were referred to as up to 14 years of age.

Eccles, J.S. The development of children ages 6 to 14. Future Child 1999, 9, 30-44.

  1. Results: 63 studies were included; however, it is needed to add information about these studies. You are asked to add:

Response:

Based on your comment #6, I have added the journal to the list. However, as you know, studies directly dealing with SB and TMD in children are extremely limited. In one recent study, only three studies were finally selected for systemic review [2]. And since it is a review in the form of inferences based on the results of SB and TMD in many adults, it is ambiguous to present the papers included in this review as a table. Please let me not miss this opportunity to present the results. Thank you for your understanding.

de Oliveira Reis L, Ribeiro RA, Martins CC, Devito KL. Association between bruxism and temporomandibular disorders in children: A systematic review and meta-analysis. Int J Paediatr Dent. 2019 Sep;29(5):585-595. doi: 10.1111/ipd.12496.

-the setting of the included studies (hospital, sleep clinic, dental clinic, university hospital clinic, etc..)

-total number of subjects involved (please summarize the total number of subjects being described in the 63 included studies)

-mean age of the subjects

-studies design (case control, cross-sectional, ect..)

-years of publication 

-main outcome

-criteria to diagnose TMD

-criteria used to diagnose SB

-country of the study

In my opinion you should provide a table with the characteristics of the studies.

  1. In order to be consistent with the title of the study being focused on children, I think it would be better to provide an image of a child teeth/mouth instead of figure 3 that represents an adult setting.

Response:

Thank you very much for your comments. I changed the figure to a child's, not an adult.

Thank you.

Thank you.

Submission Date

01 September 2022

Date of this review

10 Sep 2022 08:59:35

Reviewer 2 Report

The presented review is biased and misleading. I found the following major flaws:

1. Firstly, Author has to decide what kind of review wants to write. I mean systematic review, narrative review, scoping review or other. It has to be clear because it defines the structure, methods, and reporting of your work.

2. Author wrote at the end of Introduction "Therefore, the
present study will discuss the etiology, diagnosis, and complications of bruxism, and the relationship between SB and temporomandibular disorders (TMDs) in children were of particular interest.". It is not a clear aim of the review. Author has to clarify the aim of the review within abstract and manuscript body or primary and secondary outcome in a case of the systematic review.

3. Methodological flaws: time frame of articles searching is 1994 and 2022. This period of time is too long. Author has to focus on last 10 years.

Furthermore, Author has to add the following inclusion criteria: original articles were definite sleep bruxism has been diagnosed (Definite sleep bruxism is based on a positive instrumental assessment, with or without a positive self-report and/or a positive clinical inspection) and TMD has been diagnosed using DC/TMD protocol. Than the risk of interpretation bias is very low.

4. Author has to use taxonomy and definitions related to sleep bruxism strictly in accordance to this article: Lobbezoo F, Ahlberg J, Raphael KG, Wetselaar P, Glaros AG, Kato T, Santiago V, Winocur E, De Laat A, De Leeuw R, Koyano K, Lavigne GJ, Svensson P, Manfredini D. International consensus on the assessment of bruxism: Report of a work in progress. J Oral Rehabil. 2018 Nov;45(11):837-844. doi: 10.1111/joor.12663.

5. Author has to use taxonomy and definitions related to TMD strictly in accordance to this article: Schiffman E, Ohrbach R, Truelove E, Look J, Anderson G, Goulet JP, List T, Svensson P, Gonzalez Y, Lobbezoo F, Michelotti A, Brooks SL, Ceusters W, Drangsholt M, Ettlin D, Gaul C, Goldberg LJ, Haythornthwaite JA, Hollender L, Jensen R, John MT, De Laat A, de Leeuw R, Maixner W, van der Meulen M, Murray GM, Nixdorf DR, Palla S, Petersson A, Pionchon P, Smith B, Visscher CM, Zakrzewska J, Dworkin SF; International RDC/TMD Consortium Network, International association for Dental Research; Orofacial Pain Special Interest Group, International Association for the Study of Pain. Diagnostic Criteria for Temporomandibular Disorders (DC/TMD) for Clinical and Research Applications: recommendations of the International RDC/TMD Consortium Network* and Orofacial Pain Special Interest Group†. J Oral Facial Pain Headache. 2014 Winter;28(1):6-27. doi: 10.11607/jop.1151.

6. Author has to thoroughly read the following four reliable and outstanding articles about relationship between sleep bruxism and TMD:

Smardz J, Martynowicz H, Michalek-Zrabkowska M, Wojakowska A, Mazur G, Winocur E, Wieckiewicz M. Sleep Bruxism and Occurrence of Temporomandibular Disorders-Related Pain: A Polysomnographic Study. Front Neurol. 2019 Mar 11;10:168. doi: 10.3389/fneur.2019.00168.

Wieckiewicz M, Smardz J, Martynowicz H, Wojakowska A, Mazur G, Winocur E. Distribution of temporomandibular disorders among sleep bruxers and non-bruxers-A polysomnographic study. J Oral Rehabil. 2020 Jul;47(7):820-826. doi: 10.1111/joor.12955.

de Oliveira Reis L, Ribeiro RA, Martins CC, Devito KL. Association between bruxism and temporomandibular disorders in children: A systematic review and meta-analysis. Int J Paediatr Dent. 2019 Sep;29(5):585-595. doi: 10.1111/ipd.12496.

Manfredini D, Lobbezoo F. Sleep bruxism and temporomandibular disorders: A scoping review of the literature. J Dent. 2021 Aug;111:103711. doi: 10.1016/j.jdent.2021.103711.

and the following article about role of mental state in TMD

FlorjaÅ„ski W, Orzeszek S. Role of mental state in temporomandibular disorders: A review of the literature. Dent Med Probl. 2021;58(1):127–133. doi:10.17219/dmp/132978

7. Reporting of the review is very weak. If Author decides that this is systematic review then the review has to be report strictly in accordance to the PRISMA 2020 Statement and use PRISMA 2020 Flow Diagram https://prisma-statement.org/

8. Title, aim of the review, and conclusions have to correspond one to each other within abstract and manuscript body.

Author Response

Reviewer 2

English language and style

( ) Extensive editing of English language and style required
(x) Moderate English changes required
( ) English language and style are fine/minor spell check required
( ) I don't feel qualified to judge about the English language and style

Is the work a significant contribution to the field?

Is the work well organized and comprehensively described?

Is the work scientifically sound and not misleading?

Are there appropriate and adequate references to related and previous work?

Is the English used correct and readable?

Comments and Suggestions for Authors

The presented review is biased and misleading. I found the following major flaws:

Response:

I deeply respect your evaluation results, and I have done my best to respond to your comments and suggestions. Modified and added parts are marked in red.

  1. Firstly, Author has to decide what kind of review wants to write. I mean systematic review, narrative review, scoping review or other. It has to be clear because it defines the structure, methods, and reporting of your work.

Response:

Thank you for your comments. I made it clear that this study is a narrative review in the sections of title, abstract, and method.

  1. Author wrote at the end of Introduction "Therefore, the present study will discuss the etiology, diagnosis, and complications of bruxism, and the relationship between SB and temporomandibular disorders (TMDs) in children were of particular interest.". It is not a clear aim of the review. Author has to clarify the aim of the review within abstract and manuscript body or primary and secondary outcome in a case of the systematic review.

Response:

Thank you very much for your comments. I have rewritten that section so that the purpose of the study is clear.

  1. Methodological flaws: time frame of articles searching is 1994 and 2022. This period of time is too long. Author has to focus on last 10 years.

Response:

As you know, studies looking at sleep bruxism and TMD in children within 10 years are too limited. Therefore, I narrowed the scope of the review from 1994 to 2022 to 1999 to 2022. I hope you understand that some of the major studies were distributed from 2002 to 2012, and one paper published in 1999 was necessarily on the list to reflect the request of reviewer 1. So, I couldn't narrow it down further.

Furthermore, Author has to add the following inclusion criteria: original articles were definite sleep bruxism has been diagnosed (Definite sleep bruxism is based on a positive instrumental assessment, with or without a positive self-report and/or a positive clinical inspection) and TMD has been diagnosed using DC/TMD protocol. Than the risk of interpretation bias is very low.

Response:  

Thanks for your comments. I have added your suggestion to the M&M section of the paper. Thank you so much.

  1. Author has to use taxonomy and definitions related to sleep bruxism strictly in accordance to this article: Lobbezoo F, Ahlberg J, Raphael KG, Wetselaar P, Glaros AG, Kato T, Santiago V, Winocur E, De Laat A, De Leeuw R, Koyano K, Lavigne GJ, Svensson P, Manfredini D. International consensus on the assessment of bruxism: Report of a work in progress. J Oral Rehabil. 2018 Nov;45(11):837-844. doi: 10.1111/joor.12663.

Response:  

Thanks for your suggestion. The paper you submitted has already been included in this review list. The definition of sleep bruxism has been more clearly revised by reflecting the points suggested by these authors.

  1. Author has to use taxonomy and definitions related to TMD strictly in accordance to this article: Schiffman E, Ohrbach R, Truelove E, Look J, Anderson G, Goulet JP, List T, Svensson P, Gonzalez Y, Lobbezoo F, Michelotti A, Brooks SL, Ceusters W, Drangsholt M, Ettlin D, Gaul C, Goldberg LJ, Haythornthwaite JA, Hollender L, Jensen R, John MT, De Laat A, de Leeuw R, Maixner W, van der Meulen M, Murray GM, Nixdorf DR, Palla S, Petersson A, Pionchon P, Smith B, Visscher CM, Zakrzewska J, Dworkin SF; International RDC/TMD Consortium Network, International association for Dental Research; Orofacial Pain Special Interest Group, International Association for the Study of Pain. Diagnostic Criteria for Temporomandibular Disorders (DC/TMD) for Clinical and Research Applications: recommendations of the International RDC/TMD Consortium Network* and Orofacial Pain Special Interest Group†. J Oral Facial Pain Headache. 2014 Winter;28(1):6-27. doi: 10.11607/jop.1151.

Response:  

Thanks for your suggestion. As you may know, it is true that RDC/TMD and DC/TMD are the most international and widely used TMD diagnostic criteria, but there are still many SCI(E)-level papers using the corresponding diagnostic criteria for TMD examination. For an overall review of SB and TMD in children, studies conducted by RDC/TMD or DC/TMD equivalent criteria were included. I am also working as a dentist and specialist for TMD and orofacial pain in Korea, and at Kyung Hee University Hospital where I work, all experts diagnose TMD based on DC/TMD. Thank you for your understanding. The paper you mention is a landmark study and has been added to this review list.

  1. Author has to thoroughly read the following four reliable and outstanding articles about relationship between sleep bruxism and TMD:

Response:  

Thank you very much for your suggestion. I have read the wonderful and monumental papers on SB and TMD that you recommended. It has been directly reflected in this review paper. Thank you

Smardz J, Martynowicz H, Michalek-Zrabkowska M, Wojakowska A, Mazur G, Winocur E, Wieckiewicz M. Sleep Bruxism and Occurrence of Temporomandibular Disorders-Related Pain: A Polysomnographic Study. Front Neurol. 2019 Mar 11;10:168. doi: 10.3389/fneur.2019.00168.

Wieckiewicz M, Smardz J, Martynowicz H, Wojakowska A, Mazur G, Winocur E. Distribution of temporomandibular disorders among sleep bruxers and non-bruxers-A polysomnographic study. J Oral Rehabil. 2020 Jul;47(7):820-826. doi: 10.1111/joor.12955.

de Oliveira Reis L, Ribeiro RA, Martins CC, Devito KL. Association between bruxism and temporomandibular disorders in children: A systematic review and meta-analysis. Int J Paediatr Dent. 2019 Sep;29(5):585-595. doi: 10.1111/ipd.12496.

Manfredini D, Lobbezoo F. Sleep bruxism and temporomandibular disorders: A scoping review of the literature. J Dent. 2021 Aug;111:103711. doi: 10.1016/j.jdent.2021.103711.

: All have been added to this review.

and the following article about role of mental state in TMD

FlorjaÅ„ski W, Orzeszek S. Role of mental state in temporomandibular disorders: A review of the literature. Dent Med Probl. 2021;58(1):127–133. doi:10.17219/dmp/132978

: This study also been added to this review.

  1. Reporting of the review is very weak. If Author decides that this is systematic review then the review has to be report strictly in accordance to the PRISMA 2020 Statement and use PRISMA 2020 Flow Diagram https://prisma-statement.org/

Response:  

Many parts have been modified or added according to the opinions of reviewers. Since there have not been many direct studies on the relationship between SB and TMD in children before, I think the explanatory power of the results is weak. However, I think these attempts and approaches should continue. Thank you.

  1. Title, aim of the review, and conclusions have to correspond one to each other within abstract and manuscript body.

Response:  

Thank you for your constructive comments. Title has been modified according to the purpose of the study. All parts you pointed out are what I did my best to correct in the given time, and I marked them in red. Thank you very much.

Submission Date

01 September 2022

Date of this review

08 Sep 2022 12:48:25

Reviewer 3 Report

Dear Author,

The present study aims to discuss the etiology, diagnosis, and complications of bruxism, and the relationship between bruxism and temporomandibular disorders (TMDs) in children.

The study is of scientific interest and in line with the aims of the journal. The author guidelines have been respected. 

However, there are some issues that should be addressed. 

Abstract

The Abstract section is too long. Please follow the instruction for authors.

  • Abstract: The abstract should be a total of about 200 words maximum. The abstract should be a single paragraph and should follow the style of structured abstracts, but without headings: 1) Background: Place the question addressed in a broad context and highlight the purpose of the study; 2) Methods: Describe briefly the main methods or treatments applied. Include any relevant preregistration numbers, and species and strains of any animals used. 3) Results: Summarize the article's main findings; and 4) Conclusion: Indicate the main conclusions or interpretations. The abstract should be an objective representation of the article: it must not contain results which are not presented and substantiated in the main text and should not exaggerate the main conclusions. (https://www.mdpi.com/journal/children/instructions)

Introduction

-       In general, bruxism is a repetitive and nonfunctional activity of masticatory muscles 45 characterized by clenching or grinding of teeth. Bruxism is largely divided into sleep and 46 awake bruxism. “ Please add references.

-       As reported in the title, this review aims to clarify the relationship between sleep bruxism and TMD. However, in the Introduction section a brief introduction on TMD.

Material and methods

-       Line 76. “PRISAM”. Correct it, please.

-       Lines 82-83. “Although bruxism has an uncertain or controversial etiology, it is considered com-82 plex and multifactorial. The etiology of bruxism requires physical, psychological, heredi-83 tary, and genetic factors.” Please, add references.

-       Lines 221-223. It could be useful to report the DC/TMD classification (see Shiffman et al 2014) discussing the paper “Rongo  et al. Diagnostic criteria for temporomandibular disorders (DC/TMD) for children and adolescents: An international Delphi study-Part 1-Development of Axis I. J Oral Rehabil. 2021 Jul;48(7):836-845. doi: 10.1111/joor.13175.” in which it is well explained the diagnosis in children.

-       Line 227. Please discuss “Ferrillo et al. Temporomandibular disorders and neck pain in primary headache patients: a retrospective machine learning study. Acta Odontol Scand. 2022 Jul 29:1-7. doi: 10.1080/00016357.2022.2105945”, reporting that TMD, neck pain and primary headache could have overlapping features.

References are well written.

Author Response

Reviewer 3

Open Review

English language and style

( ) Extensive editing of English language and style required
( ) Moderate English changes required
(x) English language and style are fine/minor spell check required
( ) I don't feel qualified to judge about the English language and style

Is the work a significant contribution to the field?

Is the work well organized and comprehensively described?

Is the work scientifically sound and not misleading?

Are there appropriate and adequate references to related and previous work?

Is the English used correct and readable?

Comments and Suggestions for Authors

Dear Author,

The present study aims to discuss the etiology, diagnosis, and complications of bruxism, and the relationship between bruxism and temporomandibular disorders (TMDs) in children.

The study is of scientific interest and in line with the aims of the journal. The author guidelines have been respected. 

However, there are some issues that should be addressed. 

Response:

I sincerely appreciate your constructive comments and suggestions. I did the best I could in the time given. Please refer to the part that has been corrected or added in red. Thank you sincerely.

Abstract

The Abstract section is too long. Please follow the instruction for authors.

  • Abstract:The abstract should be a total of about 200 words maximum. The abstract should be a single paragraph and should follow the style of structured abstracts, but without headings: 1) Background: Place the question addressed in a broad context and highlight the purpose of the study; 2) Methods: Describe briefly the main methods or treatments applied. Include any relevant preregistration numbers, and species and strains of any animals used. 3) Results: Summarize the article's main findings; and 4) Conclusion: Indicate the main conclusions or interpretations. The abstract should be an objective representation of the article: it must not contain results which are not presented and substantiated in the main text and should not exaggerate the main conclusions. (https://www.mdpi.com/journal/children/instructions)

Response:

Thanks for your warm and accurate comments. Abstract has been completely modified based on your advice.

Introduction

-       “In general, bruxism is a repetitive and nonfunctional activity of masticatory muscles 45 characterized by clenching or grinding of teeth. Bruxism is largely divided into sleep and 46 awake bruxism. “ Please add references.

Response:

Thanks for your warm and accurate comments. I added 13 references to match the content. thank you. Reference have also been added to the part you pointed out.

-       As reported in the title, this review aims to clarify the relationship between sleep bruxism and TMD. However, in the Introduction section a brief introduction on TMD.

Response:

The Introduction section has been clarified in many parts. I am deeply grateful for your kind and wise advice and suggestions.

Material and methods

-       Line 76. “PRISAM”. Correct it, please.

Response:

Thank you for your comments. Thanks. However, it was judged that I did a narrative review rather than a systemic review, so I deleted this part and modified the title and other parts. Thank you.

-       Lines 82-83. “Although bruxism has an uncertain or controversial etiology, it is considered com-82 plex and multifactorial. The etiology of bruxism requires physical, psychological, heredi-83 tary, and genetic factors.” Please, add references.

Response:

A reference has been added to that section. Thanks for the constructive comments.

-       Lines 221-223. It could be useful to report the DC/TMD classification (see Shiffman et al 2014) discussing the paper “Rongo  et al. Diagnostic criteria for temporomandibular disorders (DC/TMD) for children and adolescents: An international Delphi study-Part 1-Development of Axis I. J Oral Rehabil. 2021 Jul;48(7):836-845. doi: 10.1111/joor.13175.” in which it is well explained the diagnosis in children.

Response:

Thank you very much for your comments. I've added two great papers you recommended to my paper. Your advice must have advanced my reviewer thesis further. Thank you so much.

-       Line 227. Please discuss “Ferrillo et al. Temporomandibular disorders and neck pain in primary headache patients: a retrospective machine learning study. Acta Odontol Scand. 2022 Jul 29:1-7. doi: 10.1080/00016357.2022.2105945”, reporting that TMD, neck pain and primary headache could have overlapping features.

Response:

This paper overlaps with my other references and does not address the direct relationship between TMD and SB. I read the papers you suggested carefully and thought deeply about the relationship between TMD, headache, and neck pain, and the use of artificial intelligence. Thank you

References are well written.

Response:

Thank you for your comments.

Submission Date

01 September 2022

Date of this review

10 Sep 2022 15:36:01

Round 2

Reviewer 2 Report

The manuscript has been correctly revised. Therefore I recommend to accept the manuscript unaltered.

Reviewer 3 Report

Authors modified the text according to the suggestions.

In my opinion, it is suitable for publication.